# Wheat Leaf Rust Effector Pt48115 Localized in the Chloroplasts and Suppressed Wheat Immunity

**DOI:** 10.3390/jof11010080

**Published:** 2025-01-20

**Authors:** Lulu Song, Liping Cui, Hao Li, Na Zhang, Wenxiang Yang

**Affiliations:** College of Plant Protection, Hebei Agricultural University, Technological Innovation Center for Biological Control of Crop Diseases and Insect Pests of Hebei Province, National Engineering Research Center for Agriculture in Northern Mountainous Areas, Baoding 071000, China; 18730909272@163.com (L.S.); 18233216561@163.com (L.C.); 18630686759@163.com (H.L.)

**Keywords:** wheat leaf rust, *Puccinia triticina*, effector protein, host and fungus interaction, pathogenicity

## Abstract

Wheat leaf rust caused by *Puccinia triticina* (*Pt*) is a prevalent disease worldwide, seriously threatening wheat production. *Pt* acquires nutrients from host cells via haustoria and secretes effector proteins to modify and regulate the expression of host disease resistance genes, thereby facilitating pathogen growth and reproduction. The study of effector proteins is of great significance for clarifying the pathogenic mechanisms of *Pt* and effective control of leaf rust. Herein, we report a wheat leaf rust candidate effector protein Pt48115 that is highly expressed in the late stages of infection during wheat–*Pt* interaction. Pt48115 contains a signal peptide with a secretory function and a transit peptide that can translocate Pt48115 to the host chloroplasts. The amino acid sequence polymorphism analysis of Pt48115 in seven different leaf rust races showed that it was highly conserved. Pt48115 inhibited cell death induced by Bcl-2-associated X protein (BAX) from mice or infestans 1 (INF1) from *Phytophthora infestans* in *Nicotiana benthamiana* and by DC3000 in wheat, and its 145–175 amino acids of the C-terminal are critical for its function. Furthermore, Pt48115 inhibited callose deposition and reactive oxygen species accumulation in the wheat cultivar Thatcher, demonstrating that it is an effector that enhances *Pt* virulence by suppressing wheat defense responses. Our findings lay a foundation for future studies on the pathogenesis of *Pt* during wheat–fungus interaction.

## 1. Introduction

Wheat leaf rust caused by *Puccinia triticina* (*Pt*) is an airborne fungal disease with extensive damage to wheat worldwide. In severe cases, it can cause a yield loss of more than 15–40% [1,2,3,4]. During the infection process, *Pt* mainly acquires nutrients from the host through haustoria and regulates the resistance of wheat to leaf rust by secreting effector proteins, thereby facilitating the occurrence of leaf rust [5]. Effectors can target and disrupt any stage of the plant immune response to facilitate microbial evasion of host immunity and play an important role in pathogen–host interaction [6]. Effector proteins are secreted into various host cell compartments, such as the cell membrane, nucleus, cytoplasm, and chloroplast, among others. For example, the effector protein Pst12806 secreted by wheat stripe rust was located in the chloroplasts and suppressed their functions [7]. Studying the function of effector proteins is of great significance for understanding the relationship between fungi and host interactions [8].

At present, genomics and transcriptome sequencing have become effective methods used to identify differentially expressed genes in host–fungi interactions at different stages. For example, comparative genomics integrated with association analysis identified 20 candidate effector genes corresponding to *Lr20* [9]. Moreover, candidate effector proteins of *Lr26*, *Lr2a*, and *Lr3ka* were identified through long-read de novo assembly and comparative genomics [10]. Transcriptome sequencing can measure gene expression levels, compare differences between different samples, and identify genes associated with specific conditions or diseases. For instance, six putative effector proteins for the leaf rust resistance gene *Lr28* were identified through transcriptome and in silico analysis, among which *Lr28* was found to bind strongly to the candidate protein c14094_g1_i1 to form a very stable complex [11]. Another recent transcriptome analysis of *Lr19*-virulent mutants identified eight *AvrLr19* candidates [12]. By screening a library of interactions between wheat stripe rust and *Yr26* gene, a disease resistance-related gene *TaRab18* encoding a specific small GTP-binding protein was obtained [13]. Bruce et al. [14] analyzed the expression profile of six leaf rust fungus-infected wheats through transcriptomic sequencing and obtained more than 532 candidate effector proteins. They found that 15 effector protein genes exhibited sequence polymorphism, which could be recognized by 11 different disease resistance genes, and it was speculated that they might be their corresponding avirulent genes [14]. Subsequently, two effector proteins, Pt3 and Pt27, which might correspond to the candidate avirulent genes of *Lr9*, *Lr24*, and *Lr26*, respectively, were identified [15]. A recent transcriptome analysis found that eight candidate effector proteins, PTTG_03570, PTTG_04779, PTTG_06910, PTTG_27471, PTTG_12441, PTTG_28324, PTTG_26499, and PTTG_26516 were candidate avirulence proteins of *Lr19* [12]. Furthermore, the avirulence genes *AvrLr15* and *AvrLr21* of wheat leaf rust were recently identified for the first time [16,17]. However, a repertoire of effector proteins is secreted during infection by *Pt* but only a few related studies have been conducted on their functional analysis. Therefore, the characteristics and functions of more effector proteins require further exploration to understand *Pt* pathogenicity.

In this study, transcriptome sequencing technology integrated with high-throughput screening methods and bioinformatics analysis was employed to screen and analyze the candidate effectors. The cDNA library of the late compatible interaction (6d) and early interactions (6 h, 12 h, 24 h) of the wheat leaf rust strain THTT and the wheat leaf rust susceptible cultivar Thatcher [18] was used for screening the effector protein. The early affinity interaction (MIQ) was used as the control group, and the affinity interaction 6d (MI6d) was used as the experimental group to analyze their differential gene expression profiles. Through the analysis of differentially expressed genes, it was found that *Pt48115* (GenBank accession number: XP_053016622.1) had a high expression level in the late interaction period. The preliminary functional analysis of Pt48115 verified that it was an effector protein localized in chloroplasts and that it suppressed the host immunity. The functional analysis of this effector protein can elucidate the molecular mechanisms by which pathogens manipulate host cells, thereby facilitating the development of wheat cultivars with novel resistance traits.

This study preliminarily clarified that Pt48115 is an effector protein of wheat leaf rust which can suppress wheat defense responses. Our findings lay a theoretical foundation for revealing the pathogenic mechanism of Pt48115 in the future.

## 2. Materials and Methods

### 2.1. Plant Materials, Pt Pathotypes and Primers

The following were used in the study: RNA-seq database (https://www.ncbi.nlm.nih.gov/bioproject/PRJNA609405) (accessed on 20 June 2023); seven physiological species of wheat leaf rust with different toxicity: 03-5-99 (PHTP), 04-15-7 (FHRT), 08-5-361-1 (THTT), 08-5-260-2 (THKT), 08-5-9-2 (KHHT), 08-5-11-1 (FHHT), 13-5-28-1 (JHKT); wheat cultivar Thatcher; *Nicotiana benthamiana*; DH5α; GV3101; and EtHAn.

### 2.2. Cloning and Bioinformatic Analysis of Pt148115

A fully diseased wheat plant inoculated with each *Pt* pathotype was used to extract the genomic DNA using the CTAB method. The coding sequence of Pt48115 was amplified from the gDNA of the 6 *Pt* pathotypes using the primer pair Pt148115-F and Pt148115-R (Appendix A). The reaction was performed as follows: 94 °C for 5 min, followed by 35 cycles of 30 s at 94 °C, 30 s at 60 °C, 1 min at 72 °C, and 10 min at 72 °C. The purified PCR amplicons were ligated into a pMD19-T vector (Takara Bio, Beijing, China), transformed into *Escherichia coli* DH5α by heat shock transformation, and correct clones, which were sequenced by Sangon Biotech (Shanghai, China) Co., Ltd. Sequence analysis was conducted using BLAST (http://www.ncbi.nlm.nih.gov/blast/, accessed on 6 January 2022), CD-search (http://www.ncbi.nlm.nih.gov/Structure/cdd/wrpsb.cgi, accessed on 6 January 2022), and ORF finder (http://www.ncbi.nlm.nih.gov/orffinder/, accessed on 6 January 2022). Motif analysis was performed using MEME FIMO 5.5.2. The presence of a signal peptide of Pt48115 was predicted by SignalP v4.1 (https://services.healthtech.dtu.dk/services/SignalP-4.1/, accessed on 6 January 2022). EffectorP v1.0 was used to predict whether the target protein is an apoplast or a cytoplasmic effector (https://effectorp.csiro.au/cgi-bin/script_p3.py, accessed on 6 January 2022). Prediction of whether the target protein has a transit peptide or not was performed by ChloroP 1.1. (https://services.healthtech.dtu.dk/services/ChloroP-1.1/, accessed on 6 January 2022). Transmembrane domain prediction was performed using TMHMM v2.0. (https://services.healthtech.dtu.dk/services/TMHMM-2.0/, accessed on 6 January 2022). Detailed information regarding the primers employed in this study can be found in Appendix A.

### 2.3. qRT-PCR Analysis

The highly virulent physiological race THTT of wheat leaf rust was inoculated on the susceptible wheat leaf rust variety Thatcher, and the wheat leaves were collected at 0 h, 6 h, 12 h, 24 h, 36 h, 48 h, 72 h, 96 h, 144 h, 192 h, and 216 h after inoculation. The TaKaRa MiniBEST Plant RNA Extraction Kit (TaKaRa) was used to extract RNA, reverse transcription was performed using EasyScript One-Step gDNA Removal and cDNA Synthesis SuperMix (AE311; Transgen, Beijing, China). Quantitative reverse transcription polymerase chain reaction was performed on a Quantstudio5 instrument (Thermo Fisher, Waltham, MA, USA). The transcription levels of *Pt48115* were measured using *Pt* Actin (Gene Accession OAV91054) as a reference gene [19,20,21]. Each sample was analyzed as three biological replicates. The reaction was performed as follows: 94 °C for 30 s, followed by 30 cycles of 5 s at 94 °C, 15 s at 60 °C, and 10 s at 72 °C. The relative expression of *Pt48115* was calculated by the 2^−ΔΔCt^ method. Standard deviations and averages were calculated from results of three independent biological replicates. Statistical significance was assessed using a Student *t*-test. The amplification efficiency for *Pt48115* and *PtActin* primer pairs was detected by the establishment of standard curves generated using a series of 10-fold dilutions of cDNAs according the reference [22].

### 2.4. Statistical Analysis

An unpaired two-sample Student’s *t*-test was conducted to analyze two independent datasets. For analyses involving more than two datasets, such as multiple control groups with a single gene group, a one-way ANOVA was applied. Ultimately, the *p*-value is computed, and if the *p*-value is below 0.05, a statistically significant difference between the groups is indicated.

### 2.5. Analysis of Different Physiological Polymorphisms of Pt48115

DNA of seven physiological races of *Pt* with different virulence (03-5-99 (PHTP), 04-15-7 (FHRT), 08-5-361-1 (THTT), 08-5-260-2 (THKT), 08-5-9-2 (KHHT), 08-5-11-1 (FHHT), 13-5-28-1 (JHKT)) was extracted and was used as the template for Pt48115 PCR amplification. The purified PCR amplicons were ligated into pMD19-T vector and transformed into DH5α. The acquired sequences were aligned with MEGA 7 for polymorphism analysis.

### 2.6. Yeast Signal Sequence Trap System

The predicted Pt48115 signal peptide sequence was amplified using specific primers (Appendix A), and the reaction was performed as follows: 94 °C for 5 min, followed by 35 cycles of 30 s at 94 °C, 30 s at 60 °C, 1 min at 72 °C, and 10 min at 72 °C. The purified PCR amplicons were cloned into the vector pSUC2T7M13ORI (pSUC2), and then transformed into the invertase-deficient yeast strain YTK12 and incubated at 30 °C [23]. The positive control plasmid pSUC2-Avr1b and the negative control plasmid pSUC2-Mg87 were expressed in defective yeast. The recombinant plasmid with signal peptide secretion function could grow on YPRAA medium and could secrete sucrase to hydrolyze sucrose into monosaccharides. Triphenyltetrazolium chloride, which becomes red and insoluble in water when monosaccharides react with TTC, was used to detect the ability of the enzyme to reduce 2,3,5-triphenyltetrazonium chloride (TTC) to insoluble red 1,3,5-triphenylformic acid (TPF) [24].

### 2.7. Functional Analysis of Transit Peptide

Specific primers were used to amplify the full-length ORF sequence of Pt48115 and the sequence of Pt48115 without the transit peptide (Appendix A). The reaction was performed as follows: 94 °C for 5 min, followed by 35 cycles of 30 s at 94 °C, 30 s at 55 °C, 1 min at 72 °C, and 10 min 72 °C. The purified PCR amplicon was cloned into vector pGR107. The recombinant plasmid was expressed in *Nicotiana benthamiana* using an *Agrobacterium* GV3101-mediated heterologous expression system, and after 48 h its subcellular localization was observed and photographed under a TI2-U inverted fluorescence microscope (Nikon, Tokyo, Japan).

### 2.8. Analysis of the Pt48115 Critical Sequence Required for Its Function

The procedure for PCR amplification using specific primers was the same as that for Section 2.5. By constructing the recombinant vector of the Pt48115 segmented sequence and pGR107, the mutant effector protein was expressed in tobacco using the heterologous expression system mediated by *Agrobacterium* GV3101 to verify whether it can still inhibit the necrosis reaction induced by INF1 to determine its functional domain. The recombinant plasmid pEDV6: Pt48115 was constructed and expressed in wheat by means of the bacterial type III secretion system mediated by *Pseudomonas fluorescens* strain (EtHAn). *Pseudomonas syringae* DC3000, pEDV6, and Pt48115-pEDV6 were infiltrated separately. pEDV6 + DC3000 and pEDV6-Pt48115 + DC3000 were infiltrated together, the infiltrated range was labeled, and the phenotype was observed after 3 days.

### 2.9. Pt48115 Inhibits Callose Deposits and ROS Accumulation in Wheat

The procedure for PCR amplification using specific primers was the same as that for Section 2.5. The gateway method was used to construct pEDV6:Pt48115, which was then transformed into an EtHAn strain. The effector protein was delivered into wheat through the bacterial type III secretion system. pEDV6:dsRED, empty strain EtHAn, and buffer MgCl_2_ were used as controls. Three biological replicates were collected from wheat samples of the treatment group and the control group and placed in a 2 mL centrifuge tube. A mixture of 95% ethanol–acetic acid = 1:1 was added into the centrifuge tube, and the wheat leaves were completely immersed. After 4–6 h, the decolorizing solution was replaced once until the leaves were transparent, the samples were then rinsed with double distilled water once, and 0.05% aniline blue dyeing solution was added and kept in the dark for 8–10 h. After staining, leaves were placed in double distilled water, and callose was observed under fluorescence microscope. The 10 fields of view for each leaf were observed, and the area of callose deposition was measured by Image-Pro Plus 6.0 software [25]. The observation of reactive oxygen species accumulation was carried out by adding 0.1% DAB staining solution to a centrifuge tube to completely immerse the wheat leaves, placing them under strong light for 8–10 h and decolorizing the stained samples in a 2 mL centrifuge tube containing a 95% ethanol–acetic acid = 1:1 mixture. The decolorizing solution was replaced every 4–6 h until the leaves were completely decolorized. The decolorized wheat was soaked and washed three times with double distilled water. Finally, the accumulation of active oxygen was observed under a microscope, and 30 infection points were observed on each leaf. Finally, the area of active oxygen was measured by Image-Pro Plus 6.0 software [25].

## 3. Results

### 3.1. Pt48115 Is a Candidate Effector Protein

We identified a candidate protein Pt48115 with a very high expression level at the late stage of infection from a transcriptome-sequenced cDNA library. To determine if this candidate protein is an effector protein, the software SignalP v6.0, TMHMM v2.0, and EffectorP v3.0 were used for screening (Appendix A). The probability of Pt48115 being a candidate effector protein was 0.912. Pt48115 is a small molecule protein of 175 amino acids, among which 14 amino acids are cysteine residues, which contain a signal peptide, a transit peptide, and FxC Motif and do not contain transmembrane domains, demonstrating that Pt48115 was a secreted effector protein.

The 528 bp fragment of *Pt48115* was obtained by PCR amplification (Appendix A). The characteristic analysis of the sequence (Appendix A) showed that Pt48115 encoded 175 amino acids, including cysteine residues, the first 21 amino acids of the N-terminal were signal peptides, and 22–81 amino acids were transport peptides (Figure 1A). It also contains FxC motifs and does not contain known domains. NCBI sequence alignment found that Pt48115 had 100% homology with PTTG-04698 (BBBD) (Figure 1B), and then AlphaFold3 was used to construct its tertiary model (Figure 1C).

### 3.2. Pt48115 Is Highly Expressed During the Late Infection Stage

The qRT-PCR amplification efficiencies of the primer pairs and the qRT-PCR amplification efficiencies of *Pt Actin* and *Pt48115* primer pairs were 99.2% and 97.6%, respectively, and the regression coefficient (R^2^) values were 0.998 and 0.996, respectively. The amplification efficiencies and regression coefficient values were within the threshold (90–110%), indicating that the primer pairs were appropriate for the qRT-PCR analysis (Appendix A). During compatible interaction between the wheat leaf rust highly virulent physiological race THTT and the susceptible wheat cultivar Thatcher, the expression level of *Pt48115* was low in the early stage and reached a peak at 144 h. After that, the expression level gradually decreased but was generally higher than that in the early stage, indicating that *Pt48115* was highly expressed in the late stage of infection (Figure 2). The expression pattern suggests that Pt48115 is a wheat leaf rust effector that may play an important role during *Pt*–wheat interaction, especially during the late stages of infection.

### 3.3. Pt48115 Protein Sequence Has No Polymorphism in Different Physiological Races

The genomic DNA of this gene was amplified using gene-specific primers from seven different *Pt* physiological races DNA as templates. The PCR products were sequenced by Shanghai Sangon Biotech of China. The total length of Pt48115 genomic DNA was 859 bp, containing four introns and five exons using multi-sequence comparison software. Moreover, there were no polymorphic sites of Pt48115 in seven different physiological races of leaf rust and no mutation, deletion, or insertion of amino acids, indicating that the effector protein Pt48115 is conserved among the seven physiological subspecies (Figure 3).

### 3.4. Pt48115 Signal Peptide Has Secretory Function

The secretion system of yeast invertase was used to verify the secretory function of the Pt48115 signal peptide sequence. Mg87, a non-secreted protein of Oryzae, was used as a negative control, while Avr1b, a secreted effector protein of *Phytophthora sojae*, was used as a positive control. YPRAA medium containing raffinose was used for screening. pSUC2-AVr1b, pSUC2-Pt48115, and pSUC2Mg87 were transferred into yeast strain YTK12. Due to the lack of a tryptophan synthesis gene, YTK12 could not grow in SD-Trp medium, while pSUC2 had a tryptophan synthesis gene and could grow in SD-Trp medium. However, it could not grow in the medium containing only raffinose. Both positive control Avr1b and effector protein Pt48115 could grow in the YPRAA medium, thus proving that the signal peptide of Pt48115 had a secretory function, while the signal peptide of the negative control had no secretory function, so it could not grow (Figure 4A). Further verification was performed using 2, 3, 5-triphenyltetrazolide chloride (TTC) reagent to detect the secreted enzyme activity, and both positive control and Pt48115 turned red, demonstrating the activity of invertase (Figure 4B).

### 3.5. Pt48115 Suppresses Programmed Cell Death in N. benthamiana and Wheat

Pt48115 was constructed onto vector pGR107, then transferred into strain GV3101 and infiltrated into tobacco. pGR107:GFP was used as a control, as demonstrated in Figure 5A. The results showed that the injection of effector proteins Pt48115 and pGR107:GFP alone cannot cause necrosis in tobacco, while the injection of infestans 1 (INF1) from *Phytophthora infestans* alone can cause necrosis, indicating that INF1 can cause a hypersensitive response (HR) in tobacco. pGR107:INF1 and pGR107:GFP were mixed in a 1:1 ratio and infiltrated into tobacco, and the phenotype was observed after 5 days. It was found that pGR107:GFP could not inhibit the INF1-induced PCD (programmed cell death), but Pt48115 could inhibit the PCD (Figure 5B). This suggests that the effector protein Pt48115 plays an important role in inhibiting host plant defense responses. Since Pt48115 inhibited PCD induced by INF1 in *N. benthamiana*, *Pseudomonas syringae* pv. tomato, DC3000 was selected as a cell death elicitor in wheat, and *Pseudomonas fluorescens* EtHAn was used to deliver the effector into wheat. When *P. syringae* DC3000 was infiltrated into wheat leaves, it induced HR in wheat, while pEDV6:Pt48115 and pEDV6 could not induce HR. HR occurred at the co-injection site of pEDV6 and DC3000, indicating that pEDV6 could not inhibit the necrosis reaction caused by DC3000, while no HR occurred at the co-injection site of pEDV6:Pt48115 and DC3000, demonstrating that pEDV6:Pt48115 could inhibit the necrosis reaction caused by DC3000 and suppressed host immunity (Figure 5C).

### 3.6. Pt48115 Is Localized in the Chloroplasts

The fusion expression vector of the Pt48115 effector protein and fluorescent protein GFP were constructed, and subcellular localization in tobacco was analyzed by a heterologous expression system. The results showed that the positive control pGR107:GFP was expressed in both the nucleus and cytoplasm. The tobacco injected with the fusion expression vector pGR107:Pt48115 was subjected to plasmolysis, and the chloroplasts were bright in the bright field and the fluorescent field, which suggested that the effector protein was localized in the chloroplast. According to the online software analysis, the protein contains a transit peptide sequence (APLSTPSAPQTCTYVYQPITTSGDPTGNQMTCRNAQSPQRFFICDQKSCEGTRKCTNCVS). The vector pGR107-Pt48115^∆CTP1^-GFP without the transit peptide was constructed and expressed in tobacco. The results showed that the effector protein was expressed in the whole cell without luminous dots (Figure 6). It was speculated that the localization of the effector protein in the chloroplast was due to the transit peptide.

### 3.7. Amino Acid Sequence 145–175 Is Critical for the Function of Pt48115

Four deletion mutants were constructed for Pt48115. The signal peptide sequence was first removed, and the effector protein was found to inhibit BAX-induced necrosis, and then the effector protein C-terminal was deleted. Then 90 amino acids (Δ22–112) and 60 amino acids (Δ22–82) were deleted from the N-terminal, and 30 amino acids (Δ145–175) were deleted from the C-terminal of the effector protein. The mutants were constructed into the recombinant plasmid pGR107-deletion-specific fragment and then transiently expressed in tobacco using the *Agrobacterium* GV3101-mediated heterologous expression system to verify whether the PCD reaction could be inhibited. The mutants with 90 and 60 amino acid deletions at the N-terminal inhibited BAX-induced necrosis, but the mutant with 30 amino deletions at the C-terminal could not inhibit BAX-induced necrosis. These results suggest that the amino acid sequence of the 145–175 position of the effector protein Pt48115 (SSPKHGSKASRASSASTSNPGGVSTWLSWN) is of great significance to its function (Figure 7).

### 3.8. Pt48115 Inhibits the Host’s Defense Response

Through microscopic observation, it can be seen that the infiltration of *P. fluorescent* EtHAn and pEDV6:dsRED can induce callose deposition in wheat, which is a typical characteristic of PTI. Therefore, the infiltration of EtHAn and pEDV6:dsRED into wheat can induce PTI. However, the buffer MgCl_2_ did not cause callose accumulation after infiltration of the wheat (Figure 8A). After Pt48115 was delivered to the wheat cultivar Thatcher through the bacterial type III secretion system, the expression of this gene inhibited the callose deposition induced by EtHAn by 85.9% compared with the control (Figure 8B), indicating that it can inhibit the primary defense response of wheat. Furthermore, after 24 h of infiltration, the physiological race THTT of *Pt* was inoculated onto the wheat cultivar Thatcher, and 24hpi samples were stained with DAB and observed under a microscope. However, pEDV6:Pt48115 significantly inhibited reactive oxygen species accumulation (Figure 8C), and the area statistics of reactive oxygen species showed a significant downward trend, indicating that effector protein Pt48115 can suppresses the host immunity (Figure 8D).

## 4. Discussion

As an obligate biotrophic fungus, wheat leaf rust solely relies on its living host for growth and reproduction. Upon detecting signals of foreign invasion, the host initiates a primary defense response at the cell membrane level [26]. The response includes the activation of the mitogen-activated protein kinase cascades, rapid phosphorylation of receptor-like cytoplasmic kinases, influx of calcium ions across plasma membrane, activation of calcium-dependent kinases, and reactive oxygen species signaling [27,28]. To evade this initial defense mechanism, the pathogen secretes effector proteins into the plant tissue, triggering a secondary immune response that is more specific and intense than pattern-triggered immunity (PTI), which is called effector-triggered immunity (ETI) [28]. This secondary response is often accompanied by PCD at the infection site—a phenomenon known as hypersensitive response. Consequently, transient expression assays are typically conducted in tobacco to evaluate whether these effector proteins can inhibit BAX/INF1-induced PCD.

We conducted an analysis using transcriptome sequencing data coupled with bioinformatics screening methods to ultimately obtain multiple candidate effector proteins. The identified candidate effector proteins were characterized by possessing N-terminal secreted signal peptides while lacking transmembrane domains and exhibiting small molecular weights. With advancements in information technology and online software tools, such as EffectorP v3.0 [29], SignalP v6.0 [30], and TMHMM v2.0 [31], can facilitate preliminary screenings for effector proteins. Pt48115 is one such protein which was found to be highly expressed during the late stages of leaf rust infection. Pt48115 has a functional signal peptide, lacks a transmembrane domain, contains cysteine, has no known domain, and includes a transit peptide, which has a chloroplast localization signal.

Transient expression assays in this study demonstrated that effector protein Pt48115 inhibited BAX/INF1-induced necrosis in tobacco. Additionally, we used *P. fluorescent* EtHAn to deliver Pt48115 into wheat leaves, and we found that it suppressed necrotic reactions induced by DC3000 and significantly reduced callose deposition in Thatcher wheat cells with an inhibition rate of 85.9%. Pt48115 also inhibited ROS accumulation, demonstrating its virulent function in counteracting host defense responses. Previous studies have reported that the wheat stripe rust effector Pst30 enhances infection and colonization by inhibiting both PTI and ETI pathways [32]. The transient expression of Hasp155 [33] in wheat mediated by the bacterial type III secretion system inhibited callose deposition induced by *P. fluorescent*. This inhibition resulted in a reduction in necrotic area and ROS accumulation associated with the avirulent stripe rust strain, and it was hypothesized that this effector enhances pathogen virulence by suppressing both PTI and ETI [33]. Pt9226 was also recently found to suppress PCD induced by both BAX in *N. benthamiana* and DC3000 in wheat [34]. In this study, the newly identified *Pt* effector Pt48115 is virulent to wheat and enhances *Pt* infection and proliferation by suppression of wheat defense responses.

The subcellular localization of effector proteins holds a crucial role in plant defense responses. The activation of R1-mediated HR demands the localization of Avr1 in the nucleus, whereas AVR1-mediated inhibition of cell death requires its localization in the cytoplasm [35]. The effector protein Pt48115 reported in this study possesses a chloroplast localization signal, within which the transit peptide is indispensable for targeting chloroplasts and can inhibit the production of plant defense signals, such as callose and ROS, to regulate the host’s defense response. Some studies have reported that chloroplast-localized effector proteins target Hsp70 through a non-cleavable transit peptide, resulting in the modification of the thylakoid structure and the inhibition of salicylic acid accumulation [36,37]. We hypothesized that Pt48115 with the chloroplast localization function inhibits chloroplast function during host infection.

Given the diversity of effector proteins, it is crucial to delineate their functional domains to elucidate their biochemical roles. The K, W, and Y domains located within the C-terminal region of oomycete Avr1b are essential for its functionality and exhibit structural similarities across other oomycetes [38]. We conducted deletion mutant analysis on the effector protein Pt48115 and found that the key domain essential for its function is located at the C-terminus. Deletion of this segment significantly impaired its ability to effectively inhibit PCD. Thus, it appears that residues at the C-terminus of amino acid sequence (SSPKHGSKASRASSASTSNPGGVSTWLSWN) are critical for the suppression of wheat immunity by Pt48115.

## 5. Conclusions

Pt48115 is a wheat leaf rust effector protein that promotes *Pt* infection and colonization by suppressing host immunity. It has a functional secretory signal peptide and a transit peptide. Pt48115 inhibited cell death induced by BAX/INF1 in tobacco and DC3000-induced cell death in wheat. This study lays a foundation for uncovering the pathogenic mechanisms of wheat leaf rust fungi and developing control strategies for wheat leaf rust. However, the precise role of the effector protein Pt48115 in chloroplasts in suppressing wheat immunity and promoting wheat leaf rust infection requires further investigation.

## Figures and Tables

**Figure 1 jof-11-00080-f001:**
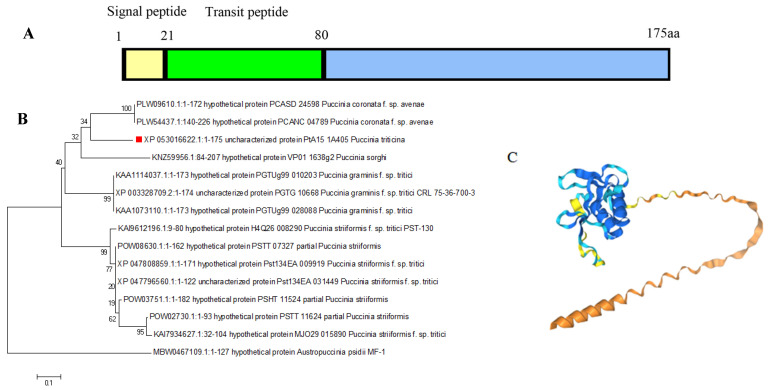
The structure and homology analysis of Pt48115. (**A**) The structural diagram of Pt48115 was made by GraphPad Prism 9.5.0. (**B**) the homology of Pt48115 was analyzed with MEGA7 software; The red dot is Pt48115. (**C**) a three-stage structural model of Pt48115.

**Figure 2 jof-11-00080-f002:**
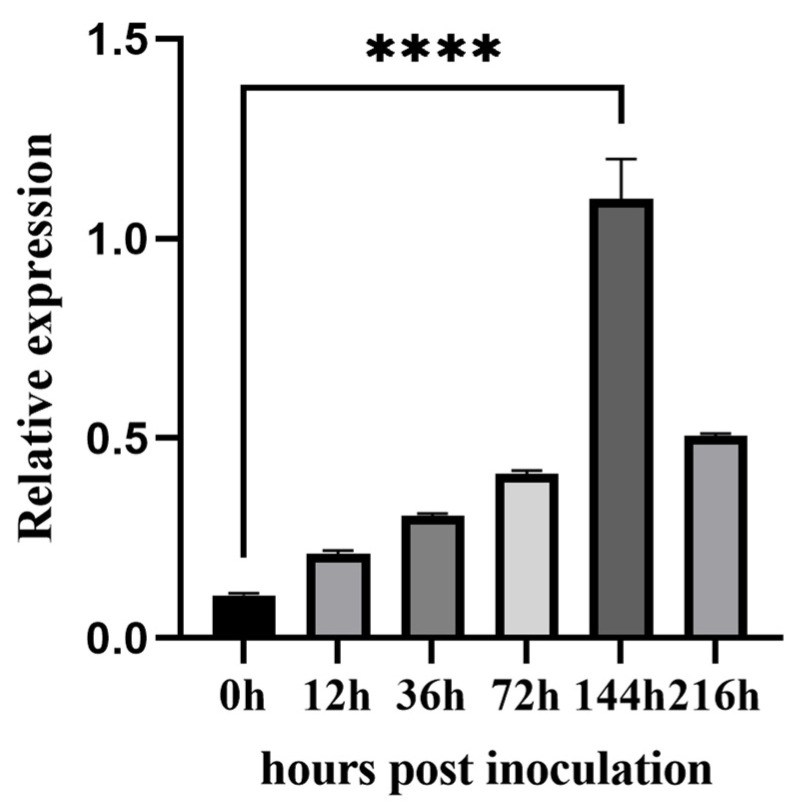
Transcription profile of *Pt48115* at different time points. *Pt48115* was highly expressed during the late stages of infection. Relative expression was calculated by the comparative 2^−ΔΔCt^ method. Standard deviation and the mean fold changes were calculated with results from three independent biological replicates. Asterisks indicate significant differences (**** *p* < 0.0001, unpaired two-tailored Student’s *t*-test).

**Figure 3 jof-11-00080-f003:**
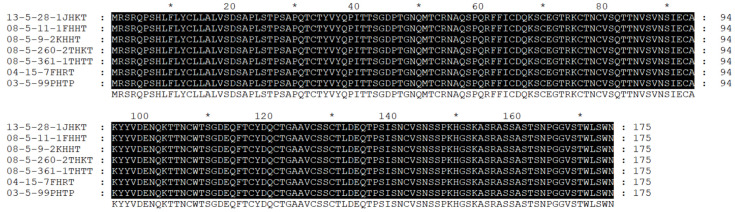
Sequence polymorphism analysis of Pt48115 among different physiological species of rusts. ‘*’ indicates positions which have a single, fully conserved residue.

**Figure 4 jof-11-00080-f004:**
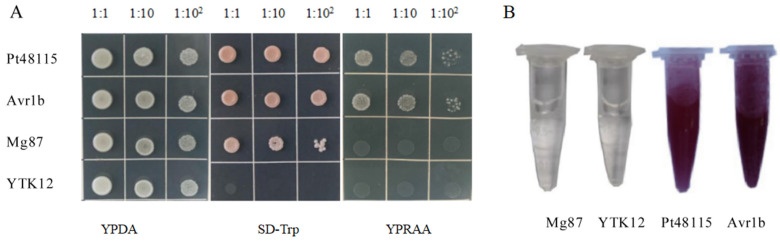
Functional validation of the Pt48115 signal peptide using the yeast invertase secretion assay. (**A**) The sequence of Pt48115 signal peptide was fused in frame to the invertase sequence in the pSUC2 vector and then transformed into the yeast strain YTK12. YTK12 carrying pSUC2-Avr1b served as positive control, and YTK12 and YTK12 carrying pSUC2-Mg87 were used as negative control. Only the yeast strains capable of secreting invertase grew on both SD-Trp and YPRAA medium. (**B**) Secreted invertase can catalyze the reduction of 2,3,5-triphenyltetrazolium chloride (TTC) to form insoluble red 1,3,5-triphenyl formazan (TPF). The presence of a red color confirms the occurrence of invertase activity.

**Figure 5 jof-11-00080-f005:**
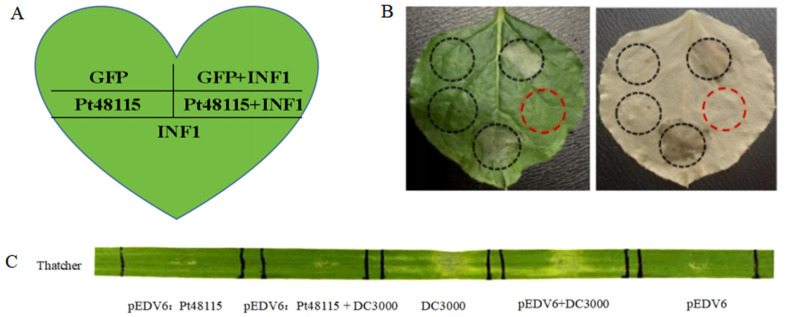
Pt48115 inhibits programmed cell death in tobacco and wheat. (**A**) schematic diagram of tobacco infiltration; (**B**) Pt48115 was transiently expressed in *Nicotiana benthamiana*, and INF1 was injected 24 h later. The same leaf was examined before (left) and after (right) staining with decolorizing solution. The red circles highlight the Pt48115 could inhibit the PCD better than the black ones. (**C**) Pt48115 delivered via *P. fluorescent* EtHAn into leaves of wheat cultivar Thatcher suppresses necrosis triggered by *P. syringae* DC3000. DC3000 and pEDV6 served as a positive and negative control, respectively.

**Figure 6 jof-11-00080-f006:**
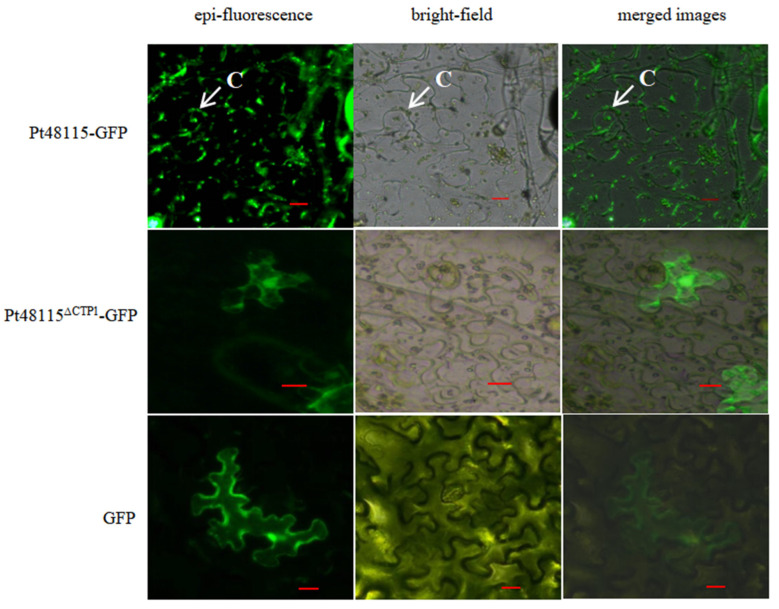
Pt48115 accumulates in the chloroplast. Leaf tissues of *Nicotiana benthamiana* transiently co-expressing Pt48115-GFP, Pt48115^ΔSP^-GFP, and GFP were examined by epifluorescence microscopy. Green means the fluorescence of the GFP. C, chloroplast. Bars = 10 μm.

**Figure 7 jof-11-00080-f007:**
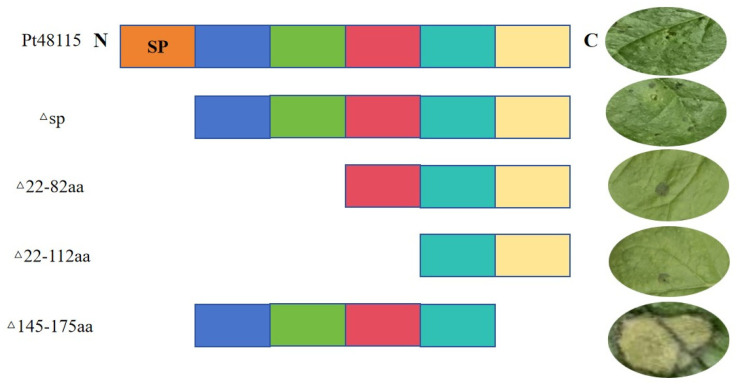
Pt48115 toxic functional domain analysis. The mutants of Pt48115 were constructed and verified on *N. benthamiana.* The amino acid sequence of the 145–175 position at the C-terminus of the effector protein Pt48115 is key for its function. SP: signal peptide; N: N terminal; C: C-terminal. The ovals represent the corresponding phenotypic observations of the deletion mutants on *N. benthamiana* leaves. Each color represents a different gene sequence. Orange, blue, green, red, mint green, yellow represent a segment of Pt48115 signal peptide amino acid sequence, 22–52, 53–82, 83–112, 113–144, 145–175 amino acid sequence.

**Figure 8 jof-11-00080-f008:**
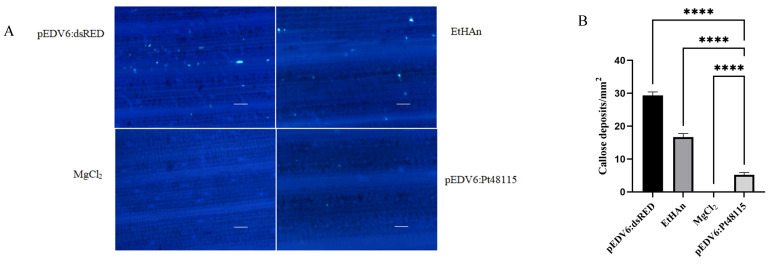
Overexpression of Pt48115 in Thatcher suppressed callose deposition and H_2_O_2_ accumulation. (**A**) Callose deposition after aniline blue staining. Wheat leaf samples were collected 24 h after infiltration of wheat cultivar Thatcher with EtHAn. After decolorization, the leaves were stained overnight with 0.05% aniline blue. pEDV6:dsRED was used as a positive control, and EtHAn and MgCl_2_ served as blank controls. Images were captured under a fluorescence microscope. Bar = 100 µm. (**B**) Statistical average number of callose deposits/mm^2^. The mean values and standard deviations were obtained from nine 1 mm^2^ areas of 3 biological replicates. Asterisks indicate significant differences (**** *p* < 0.0001, one-way ANOVA). (**C**) Wheat leaves were infiltrated with pEDV6:dsRED and pEDV6:Pt48115 and 24 h later were inoculated with the virulent *Pt* isolate THTT. pEDV6:dsRED was used as a positive control. Wheat leaf samples were collected 24 h post-inoculation with THTT. After DAB (1 mg/mL) staining, the accumulation of H_2_O_2_ was observed under a microscope. Images were captured under a microscope. Bar = 100 µm. (**D**) Statistics of H_2_O_2_ accumulation area/mm^2^ Asterisks indicate significant differences (**** *p* < 0.0001, unpaired two-tailored Student’s *t*-test).

## Data Availability

The data presented in the study are available on request from the corresponding authors due to privacy.

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
