# Peer review of "Wheat Leaf Rust Effector Pt48115 Localized in the Chloroplasts and Suppressed Wheat Immunity"

_jof, 2025, doi:10.3390/jof11010080_

Round 1

Reviewer 1 Report

The authors in their work revealed that Pt48115 is a wheat leaf rust effector protein that can inhibit the cell death program in tobacco in wheat. The authors demonstrated this using bioinfotmatical approach, overexpression assay and GFP assay in terms of protein localisation, and yeast invertase secretion. The conclusions are supported by the obtained results, the illustrations and charts are clear. 

Gene symbols in Introduction should be in italics, e.g. Lr26, Lr2a, and Lr3k (line 48), gene TaRab18 (line 50), and Yr26 gene (line 50), genes of Lr9, Lr24, and Lr26 (line 57), etc. 

Author Response

Thank you for your suggestions. All the raised issues have been addressed in the revised version of the manuscript as below.

Point 1: Gene symbols in Introduction should be in italics, e.g. Lr26, Lr2a, and Lr3k (line 48), gene TaRab18 (line 50), and Yr26 gene (line 50), genes of Lr9, Lr24, and Lr26 (line 57), etc. 

Response

All gene symbols have been italicized as per suggestion. (Line 44, Line 45, Line 49, Line 50, Line 51, Line 52, Line 53, Line 60 and Line 64)

Reviewer 2 Report

see the attached comments

see the attached comments

Author Response

Point 1: The expression profile of the effector did not align with the timeline of the host defense response; upregulation of the effector was later during the interaction, and host PTI and ETI occurred at an early point, could be within an hour post interaction. The timing of host defense is crucial to distinguish between resistant and susceptible hosts. The paper did not give any explanation for this difference.

Response

This has been revised. The expression analysis showed that Pt48115 is a wheat leaf rust effector that may play an important role during Pt-wheat interaction especially during the late stages of infection. (Line 221 – Line 223)

Point 2: The authors claimed that Pt48115 is toxic to the hosts. However, there is no evidence that Pt48115 is poison to the hosts. No results show that the effector does harm the host without the pathogens.

Response

The tone of some conclusions has been lowered as per suggestion in Line 284; Line 342 – Line 343; Line 415 – Line 416, and 424.

Point 3: The title is inappropriate because there is no evidence that Pt48115 is toxic to the hosts based on the definition of “toxic”. The correct title is “Wheat leaf rust effector Pt48115 localized in the chloroplasts and suppressed the host immunity”.

Response

This has been revised (Line 2 – Line 3)

Point 4: Misusage of terms or unclear description, for example, line 213, “… and each DNA fragment was sequenced to obtain at least four biological repeats.” It is unclear what the biological repeats mean here; was the DNA amplified from four isolates of the same race?

Response

This has been corrected. (Line 231– Line 233)

Point 5: Problems with words of choice, for example, line 214, “The total length of Pt48115 genome was 859 bp,…” It should be changed to “genomic DNA” or “gene”.

Response

This has been modified (Line 233).

Point 6: Lack of necessary information/description of non-commonly used abbreviations; for example, in line 253, when INF1 first appeared, there is no explanation of what “INF1” stands for.

Response

Explanation was given  (Line 20, Line 269-Line 270).

Point 7: Low resolution of Figure 1b.

Response

A figure with high resolution has been added

Point 8: Incorrect writing of species name, for example, when Pseudomonas fluorescens or the species of the same genus appear again, it should be presented as P. fluorescent and Pseudomonas syringae as P. syringae.

Response:

All species names have been corrected (Line 278, Line 289, Line 290, Line 330, Line 382, Line 390)

Reviewer 3 Report

The manuscript “Wheat leaf rust effector Pt48115 localized in the chloroplasts has toxic effects on wheat” presented the research conducted to describe one more effector of Puccinia triticina (Pt) the causal agent of wheat leaf rust, that poses a significant threat to global wheat production. The authors reported the candidate effector protein Pt48115, which seems to be responsible for the suppression of wheat defense responses and enhances pathogen virulence.

The importance of this study lies in its contribution to understanding the pathogenic mechanisms of Puccinia triticina (Pt), which can guide the development of novel disease management strategies, ultimately helping to secure wheat yields and global food security.

The study was well-designed, and in most cases, the authors effectively discussed their findings within the context of existing knowledge, providing a comprehensive understanding of the Pt48115’s putative role in wheat leaf rust pathogenesis.

Throughout the manuscript, several notes were added to suggest modifications that should be addressed before publication. Additionally, the authors should justify their choice of actin as the reference gene, particularly since relying on a single reference gene is insufficient to ensure the accuracy and reliability of qPCR gene expression analysis.

I recommend that the authors include at least one additional reference gene, as supported by previous studies, to enhance the accuracy and reliability of their qPCR gene expression analysis.

Detail comments included in the attached file.

Author Response

Thank you for your suggestions. All the raised issues have been addressed in the revised version of the manuscript as below.

Point 1: Furthermore, Pt48115 could inhibit callose deposition and reactive oxygen species accumulation in wheat cultivar Thatcher, demonstrating that it is an effector that plays a key role in suppressing wheat defense responses, thereby enhancing pathogen virulence. Please rewrite this sentence!

Response

This has been revised (Line 21 – Line 24)

Point 2:At present, genomics and transcriptome sequencing have emerged as effective ways of mining information about pathogenic and disease-resistance genes.  Please rewrite this sentence!

 Response

This has been revised. (Line 41 – Line 43)

Point 3: For instance, Wu et al. identified candidate effector proteins of Lr26, Lr2a, and Lr3ka through genome resequencing and comparative genomics. Authors are writing about transcriptomic and refer and example of genomic.    

Response

This has been revised. Both genomic and transcriptome sequencing literature has been added (Line 43 – Line 46, and Line 48 – Line 52)

Point 4: The functional analysis of this effector protein can reveal the molecular mechanism by which pathogens manipulate host cells, which is helpful for breeding crops with new resistance traits. Please rewrite.

Response

This has been revised (Line 79 – Line 82)

Point 5: Thatcher

Response

This has been revised (Line 91)

Point 6: The authors should justify their choice of actin as the reference gene, particularly since relying on a single reference gene is insufficient to ensure the accuracy and reliability of qPCR gene expression analysis. I recommend that the authors include at least one additional reference gene, as supported by previous studies, to enhance the accuracy and reliability of their qPCR gene expression analysis.

Response

This has been revised. The references using the ACT1 gene as an internal reference are added (Line 126, Line 489 – Line 495).

ACT1 is a commonly used reference gene for data normalization and often referred to as a House keeping gene due to its assumed stable expression across a wide range of conditions [1-3]. Furthermore, the studies of wheat rust effectors have been using a single reference gene for data normalization and for wheat leaf rust effectors  the mostly used reference gene is the PtActin [4-7], that’s why we chose it for our study.

Point 7: physiological interspecific polymorphisms

Response

This has been revised (Line 132)

Point 8: Figure1: Captions should be self-explanatory, so a reference to the software used for clustering the sequence homology, the structure, and the predicted 3D model should be included in the caption. Furthermore, some of these bioinformatics tools were not mentioned along with M&M.

Response

This has been revised. The homology of Pt48115 was analyzed with the MEGA7 software. The structural diagram of Pt48115 was made by Graphpad Prism 9.5.0. (Line 213 – Line 215)

Point 9: During compatible interaction between the wheat leaf rust highly virulent physiological race THTT and the susceptible leaf rust variety Thatcher

Response

This has been revised (Line 218)

Point 10: As previously referred, the statistical analysis used should be included in the caption. Please include in the x axis all time points. The data points do not follow the same interval.

Response

Statistical analysis had been added in the caption and all time points were modified accordingly. (Line 228 – Line 229)

Point 11: Figure 5B:It is not clear from this figure what the authors want to show or what the 2 images are.

Response

Description has been given in the Figure legend. (Line 287 – Line 291)

Point 12: Please indicate the H2O2 accumulation by arrows in 8C figure.

Response

H2O2 accumulation has been indicated by arrows in Figure 8C

Point 13: The word symbiosis relies on the assumption that both organisms are benefiting with the interaction. This is not the case in plant-rust interactions. So the authors should reformulate this sentence!

Response

This has been revised (Line 358 – Line 359)

Point 14: It is not mandatory that callose deposition occurs in the early stages of the infection... please reformulate this sentence.

Response

The sentence has been reformulated (Line 360 – Line 363)

Point 15: The authors does not report the results of the blast analysis of Pt48115 so this sentence should be removed, or the results should include the blast analysis

Response

The sentence has been removed

References

  1. Teste, M.A.; Duquenne, M.; François, J.M.; Parrou, J.L. Validation of reference genes for quantitative expression analysis by real-time RT-PCR in Saccharomyces cerevisiae. BMC Mol. Biol. 2009, 10, 99, doi:10.1186/1471-2199-10-99.
  2. Zhu, X.; Yuan, M.; Shakeel, M.; Zhang, Y.; Wang, S.; Wang, X.; Zhan, S.; Kang, T.; Li, J. Selection and evaluation of reference genes for expression analysis using qRT-PCR in the beet armyworm Spodoptera exigua (Hübner) (Lepidoptera: Noctuidae). PLoS One 2014, 9, e84730, doi:10.1371/journal.pone.0084730.
  3. Ji, Y.; Tu, P.; Wang, K.; Gao, F.; Yang, W.; Zhu, Y.; Li, S. Defining reference genes for quantitative real-time PCR analysis of anther development in rice. Acta Biochim. Biophys. Sin. 2014, 46, 305-312, doi:10.1093/abbs/gmu002.
  4. Zhao, S.; Shang, X.; Bi, W.; Yu, X.; Liu, D.; Kang, Z.; Wang, X.; Wang, X. Genome-Wide Identification of Effector Candidates With Conserved Motifs From the Wheat Leaf Rust Fungus Puccinia triticina. Front Microbiol 2020, 11, 1188, doi:10.3389/fmicb.2020.01188.
  5. Chang, J.; Mapuranga, J.; Li, R.; Zhang, Y.; Shi, J.; Yan, H.; Yang, W. Wheat leaf rust fungus effector protein Pt1641 is avirulent to TcLr1. Plants 2024, 13, 2255, doi:10.3390/plants13162255.
  6. Chang, J.; Mapuranga, J.; Wang, X.; Dong, H.; Li, R.; Zhang, Y.; Li, H.; Shi, J.; Yang, W. A thaumatin-like effector protein suppresses the rust resistance of wheat and promotes the pathogenicity of Puccinia triticina by targeting TaRCA. New Phytologist 2024, n/a, doi:10.1111/nph.20142.
  7. Wang, B.; Chang, J.; Mapuranga, J.; Zhao, C.; Wu, Y.; Qi, Y.; Yuan, S.; Zhang, N.; Yang, W. Effector Pt9226 from Puccinia triticina Presents a Virulence Role in Wheat Line TcLr15. Microorganisms 2024, 12, doi:10.3390/microorganisms12081723.

Reviewer 4 Report

This is a well-designed study that explores the function of a fungal protein as a putative effector protein. The study was conducted with the whole open reading frame and also with a truncated version of the protein. The latter to assess the function of particular domains. Overall, the results are sound but genetic modifications in the fungal cell (deletion/overexpression) are required to have a solid conclusion. I am not suggesting to include these data in the study but to lower the tone of some conclusions.

I have the following points for manuscript improvement:

In the expression analyses, please include references supporting the use of the ACT1 genes as a constitutive gene useful for data normalization. In the same point, please include the amplification efficiency of all of the primer pairs used in the study.

There is not a section describing the statistical analysis applied to the data, please amend this.

Figure 2, there is no meaning for asterisks.

Figure 7, there is no description of what is observed in the ovals, and these are too small.

Figure 8 and Figure legend. Both need improvement for clarity.

This is a well-designed study that explores the function of a fungal protein as a putative effector protein. The study was conducted with the whole open reading frame and also with a truncated version of the protein. The latter to assess the function of particular domains. Overall, the results are sound but genetic modifications in the fungal cell (deletion/overexpression) are required to have a solid conclusion. I am not suggesting to include these data in the study but to lower the tone of some conclusions.

I have the following points for manuscript improvement:

In the expression analyses, please include references supporting the use of the ACT1 genes as a constitutive gene useful for data normalization. In the same point, please include the amplification efficiency of all of the primer pairs used in the study.

There is not a section describing the statistical analysis applied to the data, please amend this.

Figure 2, there is no meaning for asterisks.

Figure 7, there is no description of what is observed in the ovals, and these are too small.

Figure 8 and Figure legend. Both need improvement for clarity.

Author Response

Point 1: This is a well-designed study that explores the function of a fungal protein as a putative effector protein. The study was conducted with the whole open reading frame and also with a truncated version of the protein. The latter to assess the function of particular domains. Overall, the results are sound but genetic modifications in the fungal cell (deletion/overexpression) are required to have a solid conclusion. I am not suggesting to include these data in the study but to lower the tone of some conclusions.

 Response

The tone of some conclusions has been lowered as per suggestion in Line 284; Line 342 – Line 343; Line 415 – Line 416, and 424.

Point 2: In the expression analyses, please include references supporting the use of the ACT1 genes as a constitutive gene useful for data normalization. In the same point, please include the amplification efficiency of all of the primer pairs used in the study. There is not a section describing the statistical analysis applied to the data, please amend this.

Response

References have been added (Line 126, Line 489 – Line 495) and the statistical analysis applied to the data has been described. (Line 228 Line 229)

Point 3: Figure 2, there is no meaning for asterisks.

Response

The meaning of the asterisks has been added (Line 229)

Point 5: Figure 7, there is no description of what is observed in the ovals, and these are too small.

Response

The figure has been modified and description added (Line 325 – Line 328).

Point 6: Figure 8 and Figure legend. Both need improvement for clarity.

Response

The figure and legend have been improved (Line 346 – Line 356).

Round 2

Reviewer 2 Report

no more suggestions

no more suggestions

Author Response

This is no more suggestions, thanks again for your professional advice

Reviewer 3 Report

The authors have addressed almost all the listed issues except for the one concerning the reference genes used. In fact, contrary to the authors' assertion, recent studies (eg. https://doi.org/10.1371/journal.pone.0309944 or https://doi.org/10.1186/s12864-015-1791-y) employ more than one reference gene. In my opinion, the study of relative gene expression should include more than one reference gene, which should be selected based on the conditions under investigation, unless previous studies support the choice made. This is not substantiated by the articles cited by the authors.

Additionally, please take note of the comment regarding Figure 8.

The authors have addressed almost all the listed issues except for the one concerning the reference genes used. In fact, contrary to the authors' assertion, recent studies (eg. https://doi.org/10.1371/journal.pone.0309944 or https://doi.org/10.1186/s12864-015-1791-y) employ more than one reference gene. In my opinion, the study of relative gene expression should include more than one reference gene, which should be selected based on the conditions under investigation, unless previous studies support the choice made. This is not substantiated by the articles cited by the authors.

Additionally, please take note of the comment regarding Figure 8.

Author Response

Point 1:  The authors have addressed almost all the listed issues except for the one concerning the reference genes used. In fact, contrary to the authors' assertion, recent studies (eg. https://doi.org/10.1371/journal.pone.0309944 or https://doi.org/10.1186/s12864-015-1791-y) employ more than one reference gene. In my opinion, the study of relative gene expression should include more than one reference gene, which should be selected based on the conditions under investigation, unless previous studies support the choice made. This is not substantiated by the articles cited by the authors. 

Response

We greatly appreciate the reviewer’s opinion for the use of two reference genes for data normalization. Using two reference genes does improve the reliability of the results, but I think using one gene can still improve the stability of the results. For example, for the effect of low temperature on Puccinia striiformis f. s. tritici, only one reference gene PstEF1 was selected as a reference gene for qRT-PCR [1]. In the functional analysis of Wrab17 gene during the interaction between wheat and Puccinia triticina, only GAPDH was selected as the reference gene [2].

Furthermore, we also calculated the amplification efficiencies of the primer pairs and the PCR amplification efficiencies of Pt Actin and Pt48115 primer pairs were 99.2% and 97.6% respectively, and the regression coefficient (R2) values were 0.998 and 0.996 respectively. The amplification efficiencies and regression coefficient values were withing the threshold (90-110%) [3], indicating that the primer pairs were appropriate for the qRT-PCR analysis – (Line 124 – Line 137).

 The references that we cited previously were to highlight the use of Pt Actin as a reference gene in the normalization of data for the expression analysis of wheat rust effectors, and we hope that you have noticed the use of a single gene for data normalization.

Point 2: Additionally, please take note of the comment regarding Figure 8.

Response

This has been revised (Line350 – Line 363)

References

  1. Ma, L.; Qiao, J.; Kong, X.; Zou, Y.; Xu, X.; Chen, X.; Hu, X. Effect of low temperature and wheat winter-hardiness on survival of Puccinia striiformisf. sp. triticiunder controlled conditions. PLoS One 2015, 10, e0130691, doi:10.1371/journal.pone.0130691.
  2. Gaoshan, Y.; Na, L.; Dong, D.; Shuaishuai, W.; Peng, L.; Shengfang, H.; Dongmei, W. Functional characterization of the Wrab17 gene in the interaction process between wheat and Puccinia triticina. Plant Physiol. Biochem. 2018, 133, 100-106, doi:10.1016/j.plaphy.2018.10.018.
  3. Pfaffl, M.W.; Tichopad, A.; Prgomet, C.; Neuvians, T.P. Determination of stable housekeeping genes, differentially regulated target genes and sample integrity: BestKeeper – Excel-based tool using pair-wise correlations. Biotechnol. Letters 2004, 26, 509-515, doi:10.1023/B:BILE.0000019559.84305.47.

Reviewer 4 Report

I thank the authors for the revised manuscript. I regret to mention that most of my original comments were not addressed. Consequently, they still stand.

The authors did not include the amplification efficiency of the primer pairs used for qPCR, which is an important aspect to assess any potential bias in the results.

The manuscript needs a subsection in the Materials and Methods section that includes all the statistical analyses used to establish whether hypotheses were accepted or rejected. This section must properly declare how the normality of data was analyzed and the particular tests used.

The ovals in Figure 7 are still too small and there is no observation of details.

Figure legends do not contain details of the statistical test used for data analysis. See for example Figure 8.

I thank the authors for the revised manuscript. I regret to mention that most of my original comments were not addressed. Consequently, they still stand.

The authors did not include the amplification efficiency of the primer pairs used for qPCR, which is an important aspect to assess any potential bias in the results.

The manuscript needs a subsection in the Materials and Methods section that includes all the statistical analyses used to establish whether hypotheses were accepted or rejected. This section must properly declare how the normality of data was analyzed and the particular tests used.

The ovals in Figure 7 are still too small and there is no observation of details.

Figure legends do not contain details of the statistical test used for data analysis. See for example Figure 8.

Author Response

Point 1:

 I thank the authors for the revised manuscript. I regret to mention that most of my original comments were not addressed. Consequently, they still stand.

The authors did not include the amplification efficiency of the primer pairs used for qPCR, which is an important aspect to assess any potential bias in the results.

Response

The amplification efficiency for Pt48115 and PtActin primer pairs was detected by the establishment of standard curves generated using a series of 10-fold dilutions of cDNAs according the reference [1]. The PCR amplification efficiencies of the primer pairs and the PCR amplification efficiencies of Pt Actin and Pt48115 primer pairs were 99.2% and 97.6% respectively, and the regression coefficient (R2) values were 0.998 and 0.996 respectively. The amplification efficiencies and regression coefficient values were withing the threshold (90-110%) according the reference [2], indicating that the primer pairs were appropriate for the qRT-PCR analysis .

Point 2:

The manuscript needs a subsection in the Materials and Methods section that includes all the statistical analyses used to establish whether hypotheses were accepted or rejected. This section must properly declare how the normality of data was analyzed and the particular tests used.

Response

This has been revised  (Line 132 – Line 134)An unpaired two-sample Student's t-test was conducted to analyze two independent datasets. For the analysis involving more than two datasets, a One-Way ANOVA was applied.

Point 3: The ovals in Figure 7 are still too small and there is no observation of details.

Response

This has been revised . 

Point 4: Figure legends do not contain details of the statistical test used for data analysis. See for example Figure 8.

Response

This has been revised (Line350-363)

References

  1. Derveaux, S.; Vandesompele, J.; Hellemans, J. How to do successful gene expression analysis using real-time PCR. Methods (San Diego, Calif.) 2010, 50, 227-230, doi:10.1016/j.ymeth.2009.11.001.
  2. Pfaffl, M.W.; Tichopad, A.; Prgomet, C.; Neuvians, T.P. Determination of stable housekeeping genes, differentially regulated target genes and sample integrity: BestKeeper – Excel-based tool using pair-wise correlations. Biotechnology Letters 2004, 26, 509-515, doi:10.1023/B:BILE.0000019559.84305.47.

Round 3

Reviewer 4 Report

Thanks for the revised version of the manuscript. I guess the author placed the text modifications in the wrong place:

The information that must be included in the STATISTICAL ANALYSES subjection was placed in lines 131-133, and this is wrong, this should be in a new subsection. In section 2.2 the authors must include the information shared in the rebuttal letter.

Thanks for the revised version of the manuscript. I guess the author placed the text modifications in the wrong place:

The information that must be included in the STATISTICAL ANALYSES subjection was placed in lines 131-133, and this is wrong, this should be in a new subsection. In section 2.2 the authors must include the information shared in the rebuttal letter.

Author Response

Point 1:

 I guess the author placed the text modifications in the wrong place:

The information that must be included in the STATISTICAL ANALYSES subjection was placed in lines 131-133, and this is wrong, this should be in a new subsection.

Response

This has been revised  (Line 135 – Line 140)  2.3. Statistical analysis: An unpaired two-sample Student's t-test was conducted to analyze two independent datasets. For analyses involving more than two datasets, such as multiple Control groups with a single gene group, a One-Way ANOVA was applied.Ultimately, the P-value is computed, and if the P-value is below 0.05, a statistically significant difference between the groups is indicated.

Point 2:

 In section 2.2 the authors must include the information shared in the rebuttal letter.

Response

This has been revised  (Line 130 – Line 134;Line 226– Line 231;)